# Rapid Syndromic Testing: A Key Strategy for Antibiotic Stewardship in ICU Patients with Pneumonia

**DOI:** 10.3390/antibiotics14050426

**Published:** 2025-04-23

**Authors:** Silvana Vulpie, Monica Licker, Oana Izmendi, Delia Muntean, Diana Lungeanu, Beatrice Sarah Zembrod, Iasmina Maria Hancu, Ovidiu Bedreag, Dorel Sandesc, Romanita Jumanca, Luminita Mirela Baditoiu

**Affiliations:** 1Doctoral School, “Victor Babes” University of Medicine and Pharmacy, 300041 Timisoara, Romania; vulpie.silvana@umft.ro (S.V.); oana.izmendi@umft.ro (O.I.); zembrodbeatrice@umft.ro (B.S.Z.); iasmina-maria.hancu@umft.ro (I.M.H.); 2Multidisciplinary Research Centre of Antimicrobial Resistance, Microbiology Department, “Victor Babes” University of Medicine and Pharmacy, 300041 Timisoara, Romania; muntean.delia@umft.ro; 3Microbiology Laboratory, “Pius Branzeu” County Clinical Emergency Hospital, 300723 Timisoara, Romania; 4Center for Modeling Biological Systems and Data Analysis, “Victor Babes” University of Medicine and Pharmacy, 300041 Timisoara, Romania; dlungeanu@umft.ro; 5Pathophyisiology Department, “Victor Babes” University of Medicine and Pharmacy, 300041 Timisoara, Romania; 6Department of Anesthesia and Intensive Care, “Victor Babes” University of Medicine and Pharmacy, 300041 Timisoara, Romania; bedreag.ovidiu@umft.ro (O.B.); sandesc.dorel@umft.ro (D.S.); 7Anesthesia and Intensive Care Unit, “Pius Branzeu” County Clinical Emergency Hospital, 300723 Timisoara, Romania; 8Romanian and Foreign Languages Department, “Victor Babes” University of Medicine and Pharmacy, 300041 Timisoara, Romania; romanita.jumanca@umft.ro; 9Epidemiology Department, “Victor Babes” University of Medicine and Pharmacy, 300041 Timisoara, Romania; baditoiu.luminita@umft.ro

**Keywords:** ventilator-associated pneumonia, antimicrobial resistance, Unyvero, standard of care

## Abstract

**Background/Objectives:** According to the European Centre for Disease Prevention and Control, improved antimicrobial stewardship programs (ASPs) combined with rapid diagnostic tests could potentially prevent thousands of deaths caused by multidrug-resistant organisms annually. This study aimed to compare the results obtained using the Unyvero system/hospital-acquired pneumonia (HPN) panel with those obtained using classic microbiological diagnostic methods to evaluate the potential of introducing this rapid diagnostic test into routine diagnosis and improving local ASPs. **Methods**: A single-center, observational, cross-sectional, analytical study was performed; it included patients admitted to the intensive care unit (ICU) with the presumptive diagnosis of community- or hospital-acquired pneumonia. One hundred non-repetitive endotracheal aspirates were collected and subjected to analysis using both methods. The concordance between the results obtained via the standard-of-care (SoC) culture and Unyvero was analyzed. **Results**: Of the results generated using Unyvero/HPN, 51% were fully concordant with those obtained via culture, 48% were partially concordant, and only 1% represented failure. It was also more efficient in identifying multiple organisms in a single sample than the SoC culture (1.32 versus 1.1 per sample). The three most common isolates identified via both methods were *Acinetobacter baumannii*, *Klebsiella pneumoniae*, and *Pseudomonas aeruginosa*. The most common resistance markers identified with Unyvero were *sul*1 (41%), *tem* and *ndm* (25%), and *kpc*, *imp*, *vim*, and *gyr*A87 (2% of results). **Conclusions**: Unyvero/HPN, if associated with appropriate diagnostic stewardship, could be used to manage critically ill patients to ensure an appropriate ASP.

## 1. Introduction

Despite major progress in medicine over the last few decades regarding the diagnosis and treatment of pneumonia, this infection remains one of the leading causes of increasing mortality and morbidity worldwide [1,2,3,4]. Whether it is community-acquired (CAP) or hospital-acquired (HAP), including the category of ventilator-associated pneumonia (VAP), it is one of the top medical conditions for which antibiotics are prescribed.

As detailed in [1], CAP is a community-acquired lung infection with a specific range of microorganisms possibly involved, including *Streptococcus pneumoniae*, respiratory viruses, *Haemophilus influenzae*, and other bacteria, such as *Mycoplasma pneumoniae* and *Legionella pneumophila*. Conversely, HAP is pneumonia that occurs in patients who were admitted to the hospital for at least 48 h and is most likely caused by organisms present in the hospital environment (methicillin-sensitive (MSSA) or methicillin-resistant (MRSA) *Staphylococcus aureus*, *Enterobacterales*, and non-fermenting Gram-negative bacilli (GNB), such as *Pseudomonas aeruginosa* and *Acinetobacter* spp.) [1,2,5,6,7]. Traditionally, this healthcare-associated infection (HAI) was further classified based on whether the patient was in an ICU and intubated at the time of infection (and, hence, at greater risk for pneumonia and poor outcomes) as ventilator-associated pneumonia (VAP), defined as pneumonia that occurs in patients from an ICU after at least 48 h from the initiation of mechanical ventilation, and non-ventilator hospital-acquired pneumonia (NV-HAP) [8].

HAP is the second most frequently diagnosed HAI after urinary tract infections, with an incidence rate ranging from 5 to more than 20 cases/1000 hospital admissions, and the most frequent cause of death secondary to an HAI, with a global mortality rate of 20–30%. Approximately one-third of HAP cases are acquired in an ICU, most being VAP. Affecting 10–25% of all patients on mechanical ventilation, VAP is the most common HAI and cause of death in critically ill patients, with an estimated global mortality rate of 20–50% [2,3,5,9]. According to several studies, one-third to one-half of deaths secondary to VAP are directly related to the infection. The highest mortality rates were reported in infections caused by non-fermenting GNB, such as *P. aeruginosa* and *Acinetobacter* spp. [2].

Many countries have included the implementation of an effective ASP in their national plans to combat antimicrobial drug resistance (AMR), considered a growing threat to patient and public health. This has been demonstrated to effectively reduce the duration of broad-spectrum antibiotic (BSA) use, improve patient outcomes, and reduce AMR and healthcare costs [10].

Therefore, it is essential to choose the most appropriate tests to allow the early identification of the causative agent and its antibiotic resistance profile. In other words, an ASP must be correlated with appropriate diagnostic stewardship to be effective [11,12].

Conventional microbiological diagnostic methods require a minimum of 48–72 h. During this time, the patient is prescribed empirical treatment, which usually consists of a BSA or an antimicrobial combination therapy, to cover the most probable organisms involved based on the guidelines, known local epidemiology, or previous colonization of patients with HAP [2,5,13]. This can lead to prolonged use of unnecessary BSAs in patients infected with susceptible pathogens or not infected by bacteria or to an incorrect treatment for patients with infections caused by a multidrug-resistant organism (MDRO) [14]. Inappropriate or prolonged use of BSAs is a well-known driver for the development and spread of AMR, significantly impacting clinical outcomes as well as public health and medical costs [15].

In this context, increasing attention has been paid in recent years to developing molecular diagnostic methods for bacterial infections, especially vital for critically ill patients admitted to ICUs [13,16].

A wide range of rapid molecular diagnostic methods are employed, such as GeneXpert technology (for screening MDROs), rapid syndromic tests (which identify organisms and resistance genes directly from the specimen), nanopore sequencing (with the rapid detection of pathogens and description of the AMR profile via polynucleotide sequencing, without needing PCR amplification), etc.

This study aimed to compare the results obtained using one of these recently developed molecular methods, namely, the Unyvero system/hospital-acquired pneumonia (HPN) panel, with those obtained using classic microbiological diagnostic methods to evaluate the potential of introducing this rapid diagnostic test into the routine diagnosis of respiratory infections in critically ill patients hospitalized in an ICU, where we are facing a high incidence of MDROs.

## 2. Results

### 2.1. Patient Characteristics

A sample of 98 patients with a clinical diagnosis of pneumonia was established in an ICU between 1 August 2021 and 31 July 2023, from which 100 respiratory specimens were collected and tested with both methods. Ninety-six patients had one single specimen, and two patients had two specimens collected. The patients’ median age was 48 years, with a slight male predominance (53.06%). All the patients received antibiotics before the sample collection. The demographic characteristics, background pathology, comorbidities, invasive devices in situ, and other risk factors are included in Table 1.

Most cases (83.67%) were healthcare-associated, with 50% being transferred from other local hospital units or neighboring counties. The fatality rate rose to 42.86% under these conditions.

### 2.2. SoC and Unyvero Concordance

One hundred non-repetitive endotracheal aspirates were collected and subjected to analysis using both methods. The concordance of the results obtained using the standard-of-care (SoC) culture and Unyvero is shown in Table 2.

In total, 110 bacterial strains were isolated using the SoC culture, while 132 were isolated using Unyvero/HPN. Table 3 presents the general detection rates of organisms in a single specimen.

The number of negative samples did not significantly differ statistically between the two methods (*p* = 0.117). There were significantly more results with one organism in the culture compared with Unyvero (*p* = 0.013), while the proportions of results with two and three isolated organisms were similar (*p* = 0.224/0.251). The rapid panel was more effective in identifying samples with four pathogens (*p* = 0.003). A single result identified five organisms in the same specimen versus zero in the case of the SoC culture (*p* = 1).

### 2.3. Pathogen Identification

The analysis of the bacterial strain identification concordance between the two methods is presented in Table 4. The overall value of the positive percent agreement (PPA) was 82.52%, and the negative percent agreement (NPA) was 96.37% (calculated after excluding species that are not included in the Unyvero panel).

The PPA was maximum (100%) for species such as *E. cloacae* complex, *E. coli*, *K. variicola*, *Proteus* spp., and *P. aeruginosa*, and minimum (50%) for *K. pneumoniae* (excluding those of 0% in the case of species that are not included in the Unyvero panel, such as *Bacillus* spp. and *P. stuartii*). The NPA ranged from 87.84% (*A. baumannii*) to 100% (*K. variicola*).

The percentages of agreement, both positive and negative, were similar for Gram-positive (GP) and Gram-negative (GN) organisms (PPA = 10/13–76.92% versus 75/90–83.33%, *p* = 0.695, respectively; NPA = 182/187–97.33% versus 1068/1110–96.22%, *p* = 0.671).

### 2.4. Antimicrobial Resistance Marker Detection

Table 5 summarizes the agreement between the results of the AMR markers detected using Unyvero and the SoC culture as phenotypic resistance data.

The positive predictive value (PPV) varied between 100% for the concordance with cephalosporin resistance markers, namely, methicillin resistance markers of *S. aureus* and the resistance phenotypes identified with the SoC, and 75% in the case of markers for resistance to macrolides and lincosamides. The percentage of genotypic markers not confirmed by the AMR phenotypes identified via microbiological culture was maximum for resistance to sulfonamides (23.81%) and minimum (0%) for *mec*A and CTX-M.

The global PPV was 90.69%, and the percentage of unconfirmed genotypic markers was 13.04%.

## 3. Discussion

In the annual surveillance report on antimicrobial resistance in Europe [17], the ECDC highlighted the high levels of AMR for several important bacterial species and groups of antimicrobial combinations. In this context, the ECDC estimated that a mixed package of interventions, including ASPs and the use of rapid diagnostic tests, could prevent thousands of deaths caused by MDROs each year and save more than EUR 1 billion a year in the EU/EEA states.

Under these circumstances, early identification of both bacteria and resistance genes to carry out the most efficient ASPs is essential. Besides their role in reducing the further emergence and spread of AMR, healthcare costs, and the adverse events associated with the unnecessary use of BSAs, one goal of ASPs is to improve the quality of care and patient outcomes by shortening the time to the most appropriate antimicrobial therapy [11,18,19].

SCJUPBT has the largest ICU in western Romania, and many patients admitted to this unit are transferred from other hospitals, either local or from neighboring counties, and are already colonized or infected with MDROs upon admission. This could partially explain the increased incidence of MDROs in this unit, which was also observed in other studies performed in the same hospital [20]. A previous study carried out in the same ICU from October 2020 to May 2021 identified a 50.42% superinfection rate in hospitalized patients with COVID-19 pneumonia [21].

In this study, 98 patients were included, from whom 100 endotracheal aspirates were collected. The patients’ median age was 48 years, with a slight male predominance. Several factors, such as prolonged hospital stay (the median for days of admission was 38) and the high percentage (50%) of patients being transferred from other healthcare facilities (one third having multiple traumas), contributed to the increased rate of HAP (83.67% of patients), most of them caused by MDROs, and also to the high rate of fatality during admission (42.86%).

Regarding the comparison between the results obtained using Unyvero/HPN and those using classical methods, the rapid test generated 51% of results fully concordant with the results obtained via the SoC culture and another 48% of partially concordant results, with only 1% failures (1 sample) of the 100 endotracheal aspirates included in this study. In 36% of the aspirates, the rapid method allowed the identification of additional bacterial species and identified 14 out of the 20 bacterial species included in the panel. Species such as *K. oxytoca*, *M. catarrhalis*, and *S. marcescens* were identified only via Unyvero, while the SoC culture identified other species that are not included in the Unyvero panel (*P. stuartii* and *Bacillus* spp., with the latter considered as a contaminant with no clinical significance) in only 7% of the samples.

The rapid method was also more efficient in identifying multiple organisms in a single sample compared with the SoC culture (1.32 versus 1.1 per sample). Considering that all the samples were collected after the initiation of antibiotic treatment, the negative culture results may have been due to either the multiplex PCR (mPCR) method identifying the remaining dead bacterial matter or the reduced loading of viable organisms. This could hypothetically occur at the onset of an infection before the culture was positive. Other authors have described the same situations in their studies [22,23,24]. They speculated the same reasons why Unyvero might have identified organisms in addition to the culture, using either the same rapid diagnostic method (Unyvero) or another diagnostic system based on PCR. Tellapragada et al. demonstrated in their initial study conducted on hospitalized patients with COVID-19 that the additional organisms identified with Unyvero could be correlated with organisms identified in the culture later on during a patientʼs hospital admission. They considered this to be Unyvero’s ability to identify a potential pathogen earlier than the classical method, allowing the clinician to efficiently treat the patient at a very early stage of the infection. Furthermore, in a follow-up study conducted on a similar group of patients, Tellapragada et al. demonstrated that the proportion of false-positive results could have been reduced from 29% to 10% if they had been correlated with the results of cultures taken 7 days before or after the date of collection of the samples subjected to Unyvero analysis [25]. The authors correlated the additional species identified with Unyvero in 12 out of 20 samples (out of a total of 69 samples included in the study) with the results of previous or subsequent cultures. However, no such correlation was found in the remaining false-positive cases. The authors speculated that the additional organisms identified could have been present in very small quantities, below the reporting level, and therefore considered colonizers, or that their presence was due to exposure to antibiotics before sample collection. The inability to clearly distinguish true pathogens from colonizers is one of the limitations of PCR-based diagnostic methods, which may lead to an undesirable effect, namely, the unnecessary prescription or escalation of antibiotic therapy in patients with this type of diagnosis [13]. However, most of these methods perform a quantitative (in the case of BioFire) or semi-quantitative (Unyvero) assessment of bacterial load. Moreover, some authors [26,27] have tried to correlate PCR signal intensity with culture growth to facilitate the correct interpretation of results in terms of clinical significance. They demonstrated that the Unyvero signal intensity in culture-positive samples was higher than in culture-negative ones.

The three most common bacterial species isolated using the SoC culture in this study were *A. baumannii* (26%), *K. pneumoniae* (24%), and *P. aeruginosa* (22%). However, the hierarchy changed using the PCR method, where the most common bacterial species were *A. baumannii* (33%), *P. aeruginosa* (25%), and *K. pneumoniae* (17%). These bacteria are known to often harbor multiple multidrug-resistant genes.

In Ganti’s study, the global PPA value that reflected the test’s ability to correctly identify patients with a certain infection was 82.52%, while the NPA, representing the ability to correctly identify patients without infections, was 96.37%. These findings have implications for managing antibiotic therapy and decreasing unnecessary antimicrobial use [28], especially in COVID-19 pneumonia cases, in which it is difficult to clinically and radiologically establish whether there is a bacterial superinfection.

Species such as *E. cloacae* complex, *E. coli*, *K. variicola*, *Proteus* spp., and *P. aeruginosa* showed an excellent PPA of 100%, and that of *A. baumannii* was 92.31%. GP had lower proportions, with 80% for *S. aureus* and 66.67% for *S. pneumoniae*. The sensitivity for *S. pneumoniae* was only 33.3% in C. Ozongwu et al.’s study, half of the value identified in this study [29]. Furthermore, other studies found lower PPA values for GP bacteria and theorized that this could be due to the different structure of their cell wall, which has a thick layer of peptidoglycan that prevents their lysis [27].

This study showed a PPA value of only 50% for *K. pneumoniae*, which is one of the main pathogens involved in HAIs in general and, hence, in HAP/VAP. Other studies showed low PPA values for the same organism, which, among others, was missed using Unyvero but found in the culture [23,26,30]. This is one of the previously reported limitations of this method. This may be because the culture may have shown small growths of *K. pneumoniae*, given the qualitative approach of the culture method, suggesting that the bacterial load was below the threshold of Unyvero identification, according to the Unyvero Application Manual—Hospitalized Pneumonia (HPN).

The most common resistance marker identified using Unyvero in this study was *sul*1 (41% of the results), followed by *tem* and *ndm* (25%), while the least frequently identified markers were *kpc*, *imp*, *vim*, and *gyr*A87 (2% of the results). The PPV, which shows the proportion of those truly infected with the resistant strain out of all test-positive subjects, was maximal for the *mec*A gene and the gene for *ctx-*M-type β-lactamase production (100%). Good values were also recorded for the gene substrates of resistance to carbapenems and class A β-lactamases. Meanwhile, the PPV was 75% for the identification of the *erm*B gene for resistance to macrolides/lincosamides. However, in those 11 samples in which the *erm*B gene was found, 2 samples had no bacteria identified, and the other 5 had only GN bacteria, which suggests that the *erm*B genes most likely belonged to some bacteria not included in the panel and forming part of the normal oropharyngeal flora. Meanwhile, *sul*1 obtained the highest value for the frequency of false resistance marker identification, followed by the one for the genetic substrate for carbapenem resistance.

Nosocomial pneumonia, with higher rates in ICUs with a surgical, oncological, or neurological profile, is feared due to its high incidence, complications, and fatality rate, depending on the AMR of the bacterial isolate [8]. The COVID-19 pandemic significantly increased the frequency of events associated with mechanical ventilation, including infectious ones. The incidence of VAP ranged from 29% in a study during the first wave of COVID-19 in Italy to 79% in a 2020 French study [23,31].

Considering the high AMR rates in our hospital, the SCJUPBT Antibiotic Prophylaxis Guideline was updated in compliance with the Stanford Antimicrobial Safety & Sustainability Program 2019 [32]. Moreover, this goal can be more easily achieved using a diagnostic tool such as Unyvero mPCR, which can provide reasonable results, especially in critically ill patients for whom every hour of inappropriate treatment can be fatal. However, proper diagnostic stewardship is crucial, which can be achieved with good multidisciplinary collaboration between the ICU and infectious disease and microbiology specialists. Due to their knowledge of clinical microbiology and the latest microbiological diagnostic tools available, microbiology specialists can contribute both to advising clinicians on choosing the most appropriate test in a given situation and at the most appropriate time and to interpreting the significance of the results obtained, thus assisting them in choosing the most appropriate management for their patients.

The novelty of this study consists of the calculation of several concordances to evaluate the potential of introducing this rapid diagnostic test into the routine diagnosis of respiratory infections in critically ill patients hospitalized in ICUs, where we are facing a high incidence of MDROs, and improving the local ASP.

However, this study has the following limitations. All the endotracheal aspirate samples were collected after the initiation of antibiotic therapy due to the severity of the clinical condition, thus increasing the probability of the Unyvero mPCR method identifying non-viable microorganisms, including traces of a previous infection. Furthermore, no data on the effectiveness of using Unyvero/HPN in the further management of antibiotic treatment were collected. This will be the future direction of our work. Moreover, some technical issues cannot be excluded (such as the samples not being vortexed properly before loading, etc.) in the case of the Unyvero-missing findings. Finally, this study was carried out in a single center.

## 4. Materials and Methods

### 4.1. Study Details

A single-center, observational, cross-sectional, analytical study was carried out by the Microbiology Department of the Clinical Laboratory of the “Pius Brânzeu” County Clinical Emergency Hospital of Timișoara (SCJUPBT) from 1 August 2021 to 31 July 2023. This study included all the patients admitted to the ICU, with the presumptive diagnosis of CAP or HAP (including VAP), from which endotracheal aspirate samples were collected and underwent molecular testing with the Unyvero/HPN panel and the culture using the routine microbiological method (considered the gold standard) simultaneously. This ICU, the largest in western Romania, has both medical and surgical profiles, with 56 beds for mixed pathology and a separate unit with 12 beds for COVID-19 patients at the time of this study.

Patients ≤ 18 years of age, those admitted to the ward for less than 24 h, and all those with other medical or surgical pathologies (but without any association with CAP, HAP, or VAP) were excluded.

Pneumonia was defined as the presence of a suggestive image in a single chest X-ray or CT scan for patients without underlying lung or cardiac pathology (or in 2 or more X-ray/CT scans in those with pre-existing lung or heart disease if comparison with previous examinations was not possible) and at least one of the following symptoms:-Fever of >38 °C without other cause;-Leukopenia (<4000 leukocytes/mm^3^) or leukocytosis (≥12,000 leukocytes/mm^3^);-New onset of purulent sputum or a change in sputum characteristics (i.e., color, smell, quantity, and consistency);-Cough or dyspnea/tachypnea;-Suggestive auscultatory signs (e.g., rales, rhonchi, and wheezing);-Deterioration of gas exchange [33].

Pneumonia was classified as CAP, an acute infection that fell within the case definition and with clinical manifestation at the time of hospital admission, hence, community-acquired, in patients who did not live in long-term care facilities nor had been hospitalized in the previous 48 h.

It was considered HAP if the clinical onset was after ≥48 h of admission (if the case came from the same medical unit) or if the patient presented to the hospital with symptoms that corresponded to the case definition and was re-admitted after <48 h upon discharge from another healthcare facility.

VAP was defined as pneumonia occurring in a patient with an invasive respiratory device in situ, even intermittently, in the 48 h preceding the onset of infection.

Variables such as patient demographic characteristics (age and sex), comorbidities, possible iatrogenic risk factors, and clinical outcomes were retrospectively collected from the electronic medical records of each patient included in this study. The data collection complied with the requirements of EU Regulation No. 679/2016 on the protection of natural persons regarding the processing of personal data.

### 4.2. Microbiological Method

The non-repetitive endotracheal aspirate samples collected from all the patients with a clinical diagnosis of CAP/HAP were simultaneously tested using the standard microbiological diagnostic method (qualitative culture of minimally contaminated specimens from the lower respiratory tract) and the mPCR-based Unyvero equipment–HPN application. A new sample from the same patient was considered non-duplicate and introduced into the database only if it was collected at least 7 days after the first one or if another microbial species or a new resistance phenotype was identified, which was completely different from the previously included one.

#### 4.2.1. Standard-of-Care (SoC) Testing

Endotracheal aspirates were collected according to the hospital’s manual for sampling, inoculated on culture media (Thermo Fisher Scientific, Wesel, Germany), and processed according to the microbiology laboratory procedures. Pathogen identification was performed using matrix-assisted laser ionization/desorption on-flight mass spectrometry (MALDI Biotyper, Bruker, Bremen, Germany), and AST was performed using VITEK^®^ 2 Compact (BioMerieux, Marcy l’Etoile, France). The AST results were interpreted according to the CLSI breakpoints.

#### 4.2.2. Testing of Study Samples Using the Unyvero/HPN Application

In this study, molecular testing was performed using the Unyvero/HPN system (Curetis GmbH, Holzgerlingen, Germany). The Unyvero/HPN application is a diagnostic system that uses a molecular approach for the syndromic testing of respiratory samples based on the rapid mPCR technology. It allows the identification, directly from patient respiratory specimens (sputum, tracheal aspirates, and bronchoalveolar lavage), of the 21 pathogens considered to be most frequently involved in the respiratory pathology of hospitalized patients, as well as of 17 resistance genes potentially responsible for the appearance of some resistance phenotypes (Table 6).

This assay allowed the results to be obtained in approximately 4.5 h.

According to the manufacturer’s recommendations, 180 μL of each sample were added into a sample tube (containing a lysis buffer), which was then introduced into a lysator for a 30 min lysis. Afterwards, the sample tube and a Master Mix tube were introduced into an HPN Unyvero cartridge for fully automated subsequent processing in an analyzer (DNA extraction, PCR, and array hybridization) in another 4 h.

### 4.3. Interpretation of Concordances

The following definitions were used to compare the two methods.

Concordance was considered if the results obtained using Unyvero and the SoC were fully identical (either negative or positive, with the same organisms). The results that reported one or more organisms identical with Unyvero and the SoC and had other organisms detected using one of the methods were considered partially concordant. If the Unyvero vs. SoC results reported completely different microorganisms, they were considered completely discordant.

The rates of true-positive (TP), true-negative (TN), false-positive (FP), and false-negative (FN) results were identified to calculate the proportions of concordant positive and negative results [34]. A positive/negative result was considered when the organism was or was not detected in that specimen.

PPA represents the ratio between the concordant positive results using microbiological culture and Unyvero and the sum of the true-positive and false-negative ones (negative with Unyvero but positive with the culture: PPA = TP/(TP + FN)). The NPA was calculated as the ratio between the concordant negative results using the microbiological culture and Unyvero and the sum of these results with the false-positive ones (positive using Unyvero but negative using the culture: NPA = TN/(TN + FP)). The relative frequency of identification of a species was estimated as the total number of bacterial isolates of the same species identified in the total number of analyzed samples.

The concordance of the resistance genotype detected using Unyvero with the phenotype identified in the AST was analyzed by calculating the PPV (the probability of a resistance marker indicated by the Unyvero panel to be confirmed using the classic method). It was obtained by reporting the number of genotypic resistance markers, confirmed by the corresponding resistance phenotype, to the total number of strains identified with that resistance genotype (PPV = TP/(TP + FP)). In addition, the proportion of genotypic resistance markers in the case of the sensitive SoC culture results (i.e., the rate of false genotypic resistance markers) was calculated using the formula FP/(FP + TN). Only the specimens for which the SoC results were available (either sensitive or resistant) were included in these calculations (namely, the PPV and FP rates).

### 4.4. Statistical Analysis

The statistical analysis was performed with the statistical software IBM SPSS v. 20 (SPSS Inc., Chicago, IL, USA) and R v. 4.3.1. (R Core Team, Vienna, Austria), including epiR v. 2.0.60. Numerical variables were expressed as medians and interquartile ranges (IQRs), and categorical variables were characterized by values and percentages. The Kolmogorov–Smirnov test was used to test the data distribution. Nominal variables were compared using the chi-squared test (Fisher’s exact test). Two-sided 95% confidence intervals (95% CIs) were estimated for frequencies, PPA, NPA, PPV, and FP rates. The statistical significance threshold value was ≤0.05.

## 5. Conclusions

The Unyvero mPCR panel enabled faster diagnosis of most pathogens isolated using the SoC culture and provided real information on resistance markers in 90% of the cases.

This study showed good PPA values, especially for the most common pathogens involved in HAP/VAP, and excellent NPA values. It also showed a good overall PPV in identifying genotypic resistance markers, with a low percentage of genotypic markers not confirmed by AMR phenotypes identified using the classical method. These results indicate that this rapid diagnostic method, if associated with appropriate diagnostic stewardship, could be used in the management of critically ill patients, especially those in the ICU, to ensure appropriate antimicrobial stewardship. It enables faster escalation to the most appropriate antibiotic therapy compared with a classical microbiological method (5 h vs. 48–72 h) and the de-escalation or discontinuation of antibiotics when the results are negative, preventing unnecessary BSA administration alongside all the associated benefits.

## Figures and Tables

**Table 1 antibiotics-14-00426-t001:** Patient demographics and other clinical data.

Patients	N = 98	[95% CI]
Female [n (%)]	46 (46.94)	[36.78–57.29]
Male [n (%)]	52 (53.06)	[42.71–63.22]
Age [median, IQR]	48.00 [28.00–65.00]	/
No. of days of hospital stay [median, IQR]	38.00 [22.00–54.00]	/
Transferred from another hospital [n (%)]	49 (50)	[39.73–60.27]
**Risk factors**		
Mechanical ventilation [n (%)]	98 (100)	[96.31–100.00]
Urinary catheter [n (%)]	98 (100)	[96.31–100.00]
CVC [n (%)]	96 (97.96)	[92.82–99.75]
Surgical wound [n (%)]	91 (92.86)	[85.84–97.08]
Vasopressors [n (%)]	90 (91.84)	[84.55–96.41]
Transfusion [n (%)]	78 (79.59)	[70.26–87.07]
Tracheostomy [n (%)]	60 (61.22)	[50.85–70.90]
Hemodialysis [n (%)]	35 (35.71)	[26.29–46.03]
Immunosuppression [n (%)]	17 (17.35)	[10.44–26.31]
Gastrostomy [n (%)]	3 (3.06)	[0.64–8.69]
Chemotherapy [n (%)]	0 (0)	/
Radiotherapy [n (%)]	0 (0)	/
GCS on admission * [median, IQR]	15.00 [15.00–3.00]	/
No. of antibiotics [median, IQR]	7.00 [4.00–10.00]	/
**Outcome**		
Discharge [n (%)]	51 (52.04)	[41.71–62.24]
Death [n (%)]	42 (42.86)	[32.90–53.25]
Transfer [n (%)]	5 (5.10)	[1.68–11.51]
**Acute pathology**		
HAP [n (%)]	82 (83.67)	[74.84–90.37]
CAP [n (%)]	16 (16.33)	[9.63–25.16]
**Background pathology**		
Polytrauma [n (%)]	36 (36.73)	[27.22–47.07]
Brain tumor [n (%)]	6 (6.12)	[2.28–12.85]
Influenza type A/B [n (%)]	6 (6.12)	[2.28–12.85]
HAP [n (%)]	6 (6.12)	[2.28–12.85]
CVA [n (%)]	4 (4.08)	[1.12–10.12]
COVID-19 [n (%)]	4 (4.08)	[1.12–10.12]
COPD [n (%)]	4 (4.08)	[1.12–10.12]
Brain aneurysm [n (%)]	3 (3.06)	[0.64–8.69]
Gastrointestinal tumor [n (%)]	3 (3.06)	[0.64–8.69]
Liver cirrhosis [n (%)]	3 (3.06)	[0.64–8.69]
Resuscitated cardiorespiratory arrest [n (%)]	3 (3.06)	[0.64–8.69]
Post-cesarean infection [n (%)]	3 (3.06)	[0.64–8.69]
Meningoencephalitis/encephalitis [n (%)]	3 (3.06)	[0.64–8.69]
Acute pancreatitis [n (%)]	2 (2.04)	[0.25–7.18]
Acute lower limb ischemia [n (%)]	2 (2.04)	[0.25–7.18]
Alcohol intoxication [n (%)]	2 (2.04)	[0.25–7.18]
Acute respiratory failure [n (%)]	2 (2.04)	[0.25–7.18]
Intestinal obstruction [n (%)]	1 (1.02)	[0.03–5.55]
Thermal burn by flame [n (%)]	1 (1.02)	[0.03–5.55]
Pleural empyema [n (%)]	1 (1.02)	[0.03–5.55]
DIC [n (%)]	1 (1.02)	[0.03–5.55]
Miastenia gravis [n (%)]	1 (1.02)	[0.03–5.55]
Pulmonary neoplasm [n (%)]	1 (1.02)	[0.03–5.55]
**Associated pathology**		
Sepsis [n (%)]	24 (24.49)	[16.36–34.21]
Diabetes [n (%)]	13 (13.27)	[7.26–21.62]
Pressure ulcers [n (%)]	8 (8.16)	[3.59–15.45]
Obesity [n (%)]	5 (5.10)	[1.68–11.51]
Other nephropathies ** [n (%)]	4 (4.08)	[1.12–10.12]
Nephrolithiasis [n (%)]	3 (3.06)	[0.64–8.69]
Pneumothorax [n (%)]	3 (3.06)	[0.64–8.69]
Psychopathy [n (%)]	3 (3.06)	[0.64–8.69]
COVID-19 [n (%)]	2 (2.04)	[0.25–7.18]
Crohn’s disease [n (%)]	2 (2.04)	[0.25–7.18]
*C. difficile* enterocolitis [n (%)]	1 (1.02)	[0.03–5.55]

**Abbreviations:** 95% CI—estimated confidence interval at a 95% confidence level; CVC—central venous catheter; GCS—Glasgow Coma Scale; HAP—hospital-acquired pneumonia; CAP—community-acquired pneumonia; CVA—cerebrovascular accident; COPD—chronic obstructive pulmonary disease; DIC—disseminated intravascular coagulation; *—specified only for 86 patients (87.76%); **—other nephropathies: acute pyelonephritis or renal tumors.

**Table 2 antibiotics-14-00426-t002:** Concordance of results obtained using the standard-of-care (SoC) culture and Unyvero.

Type of Results	N of Specimens, Total = 100
All concordant results (either negative or the same microorganisms) [n (%)]	51 (51.00)
Partially concordant [n (%)]	48 (48.00)
Unyvero miss, SoC finding [n (%)]	19 (19.00)
Unyvero finding, SoC miss [n (%)]	24 (24.00)
Some misses/findings both with Unyvero and SoC [n (%)]	5 (5.00)
Completely discordant results [n (%)]	1 (1.00)

**Table 3 antibiotics-14-00426-t003:** Number of microorganisms detected: Unyvero and the standard-of-care (SoC) culture results.

Results	N of Specimens, Total = 100	
	Unyvero	SoC
Negative [n (%)]	34 (34.00)	23 (23.00)
Positive [n (%)]	66 (66.00)	77 (77.00)
1 bacterial strain [n (%)]	30 (30.00)	48 (48.00)
2 bacterial strains [n (%)]	17 (17.00)	25 (25.00)
3 bacterial strains [n (%)]	9 (9.00)	4 (4.00)
4 bacterial strains [n (%)]	9 (9.00)	0 (0)
5 bacterial strains [n (%)]	1 (1.00)	0 (0)

**Table 4 antibiotics-14-00426-t004:** Unyvero results in microorganism identification compared with the standard-of-care (SoC) culture results (N = 100 specimens).

Species	Relative Frequency Based on the SoC [95% CI]	Relative Frequency Based on Unyvero [95% CI]	Positive with Unyvero and the SoC/Positive with the SoC PPA [95% CI]	Negative with Unyvero and the SoC/Negative with the SoC NPA [95% CI]
*Acinetobacter baumannii*	26/100	33/100	24/26	65/74
	26% [17.74–35.73]	33% [23.92–43.12]	92.31% [74.87–99.05]	87.84% [78.16–94.29]
*Bacillus* species *	2/100 2% [0.24–7.04]	0/100 0% [0–3.62]	0/2 0% [0–53.71]	98/98 100% [96.31–100]
*Citrobacter freundii*	0/100 0% [0–3.62]	1/100 1% [0.03–5.45]	/	99/100 99% [94.55–99.97]
*Enterobacter cloacae* complex	2/100 2% [0.24–7.04]	5/100 5% [1.64–11.28]	2/2 100% [15.81–100]	95/98 96.94% [91.31–99.36]
*Escherichia coli*	4/100 4% [1.10–9.93]	7/100 7% [2.86–13.89]	4/4 100% [39.76–100]	93/96 96.88% [91.14–99.35]
*Klebsiella oxytoca*	0/100 0% [0–3.62]	4/100 4% [1.10–9.93]	/	96/100 96% [90.07–98.90]
*Klebsiella pneumoniae*	24/100 24% [16.02–33.57]	17/100 17% [10.23–25.82]	12/24 50% [29.12–70.88]	71/76 93.42% [85.31–97.83]
*Klebsiella variicola*	2/100 2% [0.24–7.04]	2/100 2% [0.24–7.04]	2/2 100% [15.81–100]	98/98 100% [96.31–100]
*Moraxella catarrhalis*	0/100 0% [0–3.62]	2/100 2% [0.24–7.04]	/	98/100 98% [92.96–99.76]
*Proteus* species	6/100 6% [2.23–12.60]	7/100 7% [2.86–13.89]	6/6 100% [54.07–100]	93/94 98.94% [94.21–99.97]
*Providencia stuartii* *	5/100 5% [1.64–11.28]	0/100 0% [0–3.62]	0/5 0% [0–53.71]	95/95 100% [96.19–100]
*Pseudomonas aeruginosa*	22/100 22% [14.33–31.39]	25/100 25% [16.88–34.66]	22/22 100% [84.56–100]	75/78 96.15% [89.17–99.20]
*Staphylococcus aureus*	10/100	11/100	8/10	87/90
	10% [4.90–17.62]	11% [5.62–18.83]	80% [44.39–97.48]	96.67% [90.57–99.31]
*Serratia marcescens*	0/100	1/100	/	99/100
	0% [0–3.62]	1% [0.03–5.45]		99% [94.55–99.97]
*Stenotrophomonas maltophilia*	4/100	13/100	3/4	86/96
	4% [1.10–9.93]	13% [7.11–21.20]	75% [19.41–99.37]	89.58% [81.68–94.89]
*Streptococcus pneumoniae*	3/100	4/100	2/3	95/97
	3% [0.62–8.52]	4% [1.10–9.93]	66.67% [9.43–99.16]	97.94% [92.75–99.75]

**Abbreviations**: 95% CI—estimated confidence interval at a 95% confidence level; NPA—negative percent agreement; PPA—positive percent agreement; SoC—standard of care; *—bacterial species not included in the Unyvero panel.

**Table 5 antibiotics-14-00426-t005:** Genotype (Unyvero resistance markers) agreement with the phenotypic standard-of-care (SoC) culture results for the samples (N = 100).

Type of Resistance	N of Samples with the SoC Culture Results	Genotype Markers	Observed Phenotypic Agreement Genotype Markers for the Resistant SoC Results/Genotype Markers with Unyvero PPV [95% CI]	Proportion of False Genotype Markers Genotype Markers for the Sensitive SoC Results/Sensitive SoC Results [95% CI]
Penicillins	69	*tem*; *shv*	29/31 93.55% [78.58–99.21]	2/11 18.18% [2.28–51.78]
Cephalosporins, generations 1, 2, 3, and 4	70	*ctx*-M	9/9 100% [66.37–100]	0/17 0% [0–19.51]
Monobactams and carbapenems	68	*kpc*; *imp*; *vim*; *ndm*; and *oxa*-23, −24/40, −48, and −58	35/38 92.10% [78.62–98.34]	4/18 22.22% [6.41–47.64]
Sulfonamides	62	*sul*1	27/32 84.38% [67.21–94.72]	5/21 23.81% [8.22–47.17]
Fluoroquinolones	77	*gyr*A83; *gyr*A87	14/16 87.50% [61.65–98.45]	2/22 9.09% [1.12–29.16]
Methicillin-resistant *Staphylococcus aureus* (MRSA)	10	*mec*A	2/2 100% [15.81–100]	0/6 0% [0–45.93]
Macrolides and lincozamides	13	*erm*B	3/4 75% [19.41–99.37]	1/5 20% [0.51–71.64]

**Abbreviations:** 95% CI—estimated confidence interval at a 95% confidence level; PPV—positive predictive value; SoC—standard of care. The following antibiotics were not included in the analysis: benzylpenicillin; amikacin; colistin; gentamicin; linezolid; minocycline; tetracycline; tigecycline; rifampicine; tobramycin; and vancomycin.

**Table 6 antibiotics-14-00426-t006:** Microorganisms and resistance markers identified using the Unyvero/HPN application (according to the Unyvero Application Manual—Hospitalized Pneumonia (HPN)).

Sample Types	Sputum, Bronchoalveolar Lavage, and Tracheal Aspirates		
Category	Organism	Antibiotic Resistance Genes	
Gram-positive bacteria	*Staphylococcus aureus*	Macrolides/lincosamides	*erm*B
	*Streptococcus pneumoniae*	Oxacillins	*mec*A and *mec*C
Enterobacteriaceae	*Citrobacter freundii*	Penicillins	*tem* and *shv*
	*Escherichia coli*	Third-generation cephalosporins	*ctx*-M
	*Enterobacter cloacae c*omplex	Carbapenems	*kpc*, *imp*, *ndm*, *oxa*-23, *oxa*-24/40, *oxa*-48, *oxa*-58, and *vim*
	*Enterobacter aerogenes*	Sulfonamides	*sul*1
	*Proteus* spp.	Fluoroquinolones	*gyr*A83 and *gyr*A87
	*Klebsiella pneumoniae*		
	*Klebsiella oxytoca*		
	*Klebsiella variicola*		
	*Serratia marcescens*		
	*Morganella morganii*		
Non-fermenting bacteria	*Moraxella catarrhalis*		
	*Pseudomonas aeruginosa*		
	*Acinetobacter baumannii* complex		
	*Stenotrophomonas maltophilia*		
	*Legionella pneumophila*		
Others/fungi	*Pneumocystis jirovecii*		
	*Haemophilus influenzae*		
	*Mycoplasma pneumoniae*		
	*Chlamydophila pneumoniae*		

## Data Availability

The original contributions presented in this study are included in this article. Further inquiries can be directed to the corresponding author.

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
