# Peer review of "Rapid Syndromic Testing: A Key Strategy for Antibiotic Stewardship in ICU Patients with Pneumonia"

_antibiotics, 2025, doi:10.3390/antibiotics14050426_

Round 1

Reviewer 1 Report

Comments and Suggestions for Authors

It is my pleasure to review the article. Every study evaluating the effectiveness of point-of-care methods for rapid microbiological diagnostics in the ICU is of great importance, and it is necessary to present the results. The methodology is adequate, but the limitation is that this study was conducted in a single center.

Author Response

Comments 1: It is my pleasure to review the article. Every study evaluating the effectiveness of point-of-care methods for rapid microbiological diagnostics in the ICU is of great importance, and it is necessary to present the results. The methodology is adequate, but the limitation is that this study was conducted in a single center.

Response 1: Thank you for pointing this out. We agree with your opinion, therefore we added this comment in the limitation section.

Reviewer 2 Report

Comments and Suggestions for Authors

The introduction and the discussion could be more concise, focusing on the key points. One issue with the study is the prior administration of antibiotics, which may complicate the results by detecting dead bacteria, as the researchers have mentioned. The Univero system's results do not show significant differences compared to routine culturing. According to the Univero system, results are available more quickly, allowing the patients to receive suitable antibiotics sooner. Clinical studies should help support the superiority of the method.

Author Response

Comment 1: The introduction and the discussion could be more concise, focusing on the key points.

Response 1: Thank you for your suggestions. We carefully revised the introduction and discussion section and removed all passages where we could consider that their absence would not affect the overall meaning of a paragraph.

Comment 2: One issue with the study is the prior administration of antibiotics, which may complicate the results by detecting dead bacteria, as the researchers have mentioned.

Response 2: Yes, indeed, this is an issue in general with all the PCR based rapid diagnostic tests and we mentioned it as one of the limitations:”Due to the severity of the clinical condition, all endotracheal aspirate samples were collected after the initiation of antibiotic therapy, thus increasing the probability of the Unyvero multiplex method to identify non-viable microorganisms, traces of a previous infection.”

Comment 3: The Unyvero system's results do not show significant differences compared to routine culturing. According to the Unyvero system, results are available more quickly, allowing the patients to receive suitable antibiotics sooner. Clinical studies should help support the superiority of the method.

Response 3: As we mentioned also in the limitation section, we did not collect data regarding the effectiveness of using Unyvero in the management of antibiotic treatment. This will be addressed in future studies:”Another limitation is that no data were collected on the effectiveness of using Unyvero tests in the further management of antibiotic treatment. This will be the future direction of our work.”

Reviewer 3 Report

Comments and Suggestions for Authors

Dear Authors,

congratulations on your valuable work. Please, find here below some suggestions in order to help to improve the quality of your paper.

  1. I recommend to use shorter and clearer title, please revise (e.g. include the type of the study you conducted)
  2. Please, improve the statistical section by explaining the methods you used to verify the normality of the distribution.
  3.  

Author Response

Comment 1: Dear Authors, congratulations on your valuable work. Please, find here below some suggestions in order to help to improve the quality of your paper.

I recommend to use shorter and clearer title, please revise (e.g. include the type of the study you conducted).

Response 1: Thank you for your suggestion. This was pointed out by another reviewer as well and we changed the title to a shorted version: ”Rapid Syndromic Testing: A Key Strategy for Antibiotic Stewardship in ICU Patients with Pneumonia” (as suggested).

Comment 2: Please, improve the statistical section by explaining the methods you used to verify the normality of the distribution.

Response 2: The sentence ”The Kolmogorov–Smirnov test was used to test data distribution” was added in the Statistical analysis section.

Reviewer 4 Report

Comments and Suggestions for Authors

Dear Authors

Greetings

Suggestions for the Article

  1. Title and Overall Clarity – The title could be more direct and accessible, emphasizing the importance of rapid syndromic testing. Something like "Rapid Syndromic Testing: A Key Strategy for Antibiotic Stewardship in ICU Patients with Pneumonia" might be more impactful.

  2. Coherence and Flow – Some sentences can be restructured for better clarity. For example:

    • “The European Centre for Disease Prevention and Control estimated that a mixed package of interventions, including more effective antimicrobial stewardship programs (ASP) and the use of rapid diagnostic tests would have the potential to prevent thousands of deaths caused by multidrug-resistant organisms each year.”

    • A more concise version: “According to the European Centre for Disease Prevention and Control, improved antimicrobial stewardship programs (ASP) combined with rapid diagnostic tests could potentially prevent thousands of deaths caused by multidrug-resistant organisms annually.”

  3. Verb Agreement and Structure – Certain sentences could be adjusted for better grammar and fluency:

    • “Several concordances of results obtained by standard of care (SoC) culture and Unyvero were calculated.”

    • Improved version: “The concordance between results obtained through standard of care (SoC) culture and Unyvero was analyzed.”

  4. Use of Technical Terms – Ensure that all scientific terms are used correctly and consistently throughout the text. If there is a clear distinction between "hospital-acquired pneumonia" (HAP) and "ventilator-associated pneumonia" (VAP), keep that nomenclature consistent.

  5. Conclusion and Clinical Implications – The discussion section could better highlight the practical clinical impacts of the findings. For example, it could reinforce how the adoption of Unyvero might directly influence mortality rates or hospitalization duration.

The English in the article is generally understandable and technically accurate, but some areas could be refined for better clarity, flow, and grammatical correctness. Here are some key points:

  • Sentence Structure: Some sentences are long and could be restructured to improve readability. For example, phrases like "The European Centre for Disease Prevention and Control estimated that a mixed package of interventions, including more effective antimicrobial stewardship programs (ASP) and the use of rapid diagnostic tests would have the potential to prevent thousands of deaths caused by multidrug-resistant organisms each year." could be more concise and direct.

  • Verb Agreement: Expressions such as "Several concordances of results obtained by standard of care (SoC) culture and Unyvero were calculated" could be adjusted for better precision, like "The concordance between results obtained through standard of care (SoC) culture and Unyvero was analyzed."

  • Use of Articles and Prepositions: Some prepositions could be refined. For example, in "infected at all with bacteria," it would be more appropriate to say "infected at all by bacteria."

  • Terminological Consistency: Ensure that all scientific and technical terms are used consistently throughout the text.

  • Key Improvements:

    1. Sentence Structure & Clarity:

      • Some sentences are quite long, making them harder to follow. Breaking them into shorter, clearer segments improves readability. Example:

        • “The European Centre for Disease Prevention and Control estimated that a mixed package of interventions, including more effective antimicrobial stewardship programs (ASP) and the use of rapid diagnostic tests would have the potential to prevent thousands of deaths caused by multidrug-resistant organisms each year.”

        • Revised: “The European Centre for Disease Prevention and Control estimated that combining improved antimicrobial stewardship programs (ASP) with rapid diagnostic tests could potentially prevent thousands of deaths caused by multidrug-resistant organisms annually.”

    2. Verb Agreement & Grammar Refinements:

      • “Several concordances of results obtained by standard of care (SoC) culture and Unyvero were calculated.”

      • Better phrasing: “The concordance between results obtained through standard of care (SoC) culture and Unyvero was analyzed.”

      • “infected at all with bacteria” → Corrected to “infected at all by bacteria.”

    3. Consistency in Scientific Terminology:

      • The terms "hospital-acquired pneumonia" (HAP) and "ventilator-associated pneumonia" (VAP) should be used consistently throughout the article. Avoid switching terminology unnecessarily.

      • Ensure "Standard of Care (SoC)" is formatted the same way every time it's used.

    4. Formatting and Flow in Results Section:

      • Tables should be formatted consistently, and if data comparisons are made, clear explanations should precede each set of results.

      • Example: The phrase “100 non-repetitive endotracheal aspirates were collected and subjected to analysis by both methods.” could be refined to “We analyzed 100 non-repetitive endotracheal aspirates using both standard of care (SoC) culture and the Unyvero system.”

    5. Conclusion Refinement:

      • The discussion on implications for antibiotic stewardship could be made stronger by explicitly stating how Unyvero's implementation improves patient outcomes.

      • Example: Instead of “Unyvero/HPN, if associated with an appropriate diagnostic stewardship, could be used in the management of critically ill patients, in order to ensure an appropriate ASP.”, a more precise version could be: “Integrating Unyvero/HPN into routine diagnostics, alongside an effective antimicrobial stewardship program (ASP), can improve antibiotic selection and patient outcomes in ICU settings.”

Kind regards

Comments on the Quality of English Language

The English could be improved to more clearly express the research.

Author Response

Comment 1: Title and Overall Clarity – The title could be more direct and accessible, emphasizing the importance of rapid syndromic testing. Something like "Rapid Syndromic Testing: A Key Strategy for Antibiotic Stewardship in ICU Patients with Pneumonia" might be more impactful.

Response 1: Thank you for your suggestion. The title was changed, as proposed.

Comment 2: Coherence and Flow – Some sentences can be restructured for better clarity. For example:

“The European Centre for Disease Prevention and Control estimated that a mixed package of interventions, including more effective antimicrobial stewardship programs (ASP) and the use of rapid diagnostic tests would have the potential to prevent thousands of deaths caused by multidrug-resistant organisms each year.”

A more concise version: “According to the European Centre for Disease Prevention and Control, improved antimicrobial stewardship programs (ASP) combined with rapid diagnostic tests could potentially prevent thousands of deaths caused by multidrug-resistant organisms annually.”

Response 2: Thank you for your suggestion. We carefully revised the manuscript throughout and restructured certain sentences for a better coherence and flow, as suggested.

Comment 3: Verb Agreement and Structure – Certain sentences could be adjusted for better grammar and fluency:

“Several concordances of results obtained by standard of care (SoC) culture and Unyvero were calculated.”

Improved version: “The concordance between results obtained through standard of care (SoC) culture and Unyvero was analyzed.”

Response 3: The manuscript underwent English language editing, in order to improve grammar and fluency.

Comment 4: Use of Technical Terms – Ensure that all scientific terms are used correctly and consistently throughout the text. If there is a clear distinction between "hospital-acquired pneumonia" (HAP) and "ventilator-associated pneumonia" (VAP), keep that nomenclature consistent.

Response 4: The manuscript was reviewed carefully throughout and the adjustments were made, where appropriate.

Comment 5: Conclusion and Clinical Implications – The discussion section could better highlight the practical clinical impacts of the findings. For example, it could reinforce how the adoption of Unyvero might directly influence mortality rates or hospitalization duration.

Response 5: Thank you for your comments. As we mentioned in the limitation section, we did not collect data regarding the effectiveness of using Unyvero in the management of antibiotic treatment. This will be addressed in future studies:”Another limitation is that no data were collected on the effectiveness of using Unyvero tests in the further management of antibiotic treatment. This will be the future direction of our work.”

Comments 6: The English in the article is generally understandable and technically accurate, but some areas could be refined for better clarity, flow, and grammatical correctness. Here are some key points:

Sentence Structure: Some sentences are long and could be restructured to improve readability. For example, phrases like "The European Centre for Disease Prevention and Control estimated that a mixed package of interventions, including more effective antimicrobial stewardship programs (ASP) and the use of rapid diagnostic tests would have the potential to prevent thousands of deaths caused by multidrug-resistant organisms each year." could be more concise and direct.

Verb Agreement: Expressions such as "Several concordances of results obtained by standard of care (SoC) culture and Unyvero were calculated" could be adjusted for better precision, like "The concordance between results obtained through standard of care (SoC) culture and Unyvero was analyzed."

Use of Articles and Prepositions: Some prepositions could be refined. For example, in "infected at all with bacteria," it would be more appropriate to say "infected at all by bacteria."

Terminological Consistency: Ensure that all scientific and technical terms are used consistently throughout the text.

Response 6: The manuscript was reviewed carefully throughout and the adjustments were made, where appropriate.

Comments 7: Key Improvements:

Sentence Structure & Clarity:

Some sentences are quite long, making them harder to follow. Breaking them into shorter, clearer segments improves readability. Example:

“The European Centre for Disease Prevention and Control estimated that a mixed package of interventions, including more effective antimicrobial stewardship programs (ASP) and the use of rapid diagnostic tests would have the potential to prevent thousands of deaths caused by multidrug-resistant organisms each year.”

Revised: “The European Centre for Disease Prevention and Control estimated that combining improved antimicrobial stewardship programs (ASP) with rapid diagnostic tests could potentially prevent thousands of deaths caused by multidrug-resistant organisms annually.”

Verb Agreement & Grammar Refinements:

“Several concordances of results obtained by standard of care (SoC) culture and Unyvero were calculated.”

Better phrasing: “The concordance between results obtained through standard of care (SoC) culture and Unyvero was analyzed.”

“infected at all with bacteria” → Corrected to “infected at all by bacteria.”

Consistency in Scientific Terminology:

The terms "hospital-acquired pneumonia" (HAP) and "ventilator-associated pneumonia" (VAP) should be used consistently throughout the article. Avoid switching terminology unnecessarily.

Ensure "Standard of Care (SoC)" is formatted the same way every time it's used.

Response 7: The manuscript was reviewed carefully throughout and the adjustments were made, where appropriate.

Comment 8: Formatting and Flow in Results Section:

Tables should be formatted consistently, and if data comparisons are made, clear explanations should precede each set of results.

Example: The phrase “100 non-repetitive endotracheal aspirates were collected and subjected to analysis by both methods.” could be refined to “We analyzed 100 non-repetitive endotracheal aspirates using both standard of care (SoC) culture and the Unyvero system.”

Response 8: Thank you for your suggestions. The tables were designed according to the author's instructions provided by the paper and each table is preceded by a short sentence explaining the content.

Comment 9: Conclusion Refinement:

The discussion on implications for antibiotic stewardship could be made stronger by explicitly stating how Unyvero's implementation improves patient outcomes.

Example: Instead of “Unyvero/HPN, if associated with an appropriate diagnostic stewardship, could be used in the management of critically ill patients, in order to ensure an appropriate ASP.”, a more precise version could be: “Integrating Unyvero/HPN into routine diagnostics, alongside an effective antimicrobial stewardship program (ASP), can improve antibiotic selection and patient outcomes in ICU settings.”

Response 9: Our study focused on the comparison between the microbiological results obtained by two methods used to analyse the endotracheal aspirate samples collected from critically ill patients with pneumonia. We did not collect data regarding the effectiveness of using Unyvero in the management of antibiotic treatment and we plan to continue our studies in this direction, to further assess the impact of Unyvero implementation on patient's outcomes.

Reviewer 5 Report

Comments and Suggestions for Authors

The manuscript entitled "Assessing the need for rapid syndromic testing to ensure a correct antibiotic stewardship strategy in patients with pneumonia hospitalized in intensive care units" compares the results obtained using Unyvero system / Hospital-acquired pneumonia (HPN) panel with those obtained using classic microbiological diagnostic methods to evaluate the potential of introducing this rapid diagnostic test in the routine diagnosis and to improve a local ASP. The study was well-designed and the manuscript was generally informative. However, some revisions are required to improve the manuscript.

  • The title should be shortened to be more concise.
  • The full name of abbreviations should be provided at first mention in all text including abstract.
  • In Table 1, Male and Female should be written instead of Sex M and Sex F, respectively.
  • The genus names of the bacteria should be abbreviated after first use.
  • No need to mention the aim of the study in Discussion.
  • The text should be read carefully to correct the typos.

Author Response

Comment 1: The manuscript entitled "Assessing the need for rapid syndromic testing to ensure a correct antibiotic stewardship strategy in patients with pneumonia hospitalized in intensive care units" compares the results obtained using Unyvero system / Hospital-acquired pneumonia (HPN) panel with those obtained using classic microbiological diagnostic methods to evaluate the potential of introducing this rapid diagnostic test in the routine diagnosis and to improve a local ASP. The study was well-designed and the manuscript was generally informative. However, some revisions are required to improve the manuscript.

The title should be shortened to be more concise.

Response 1: Thank you for your suggestion. The title was changed to”Rapid Syndromic Testing: A Key Strategy for Antibiotic Stewardship in ICU Patients with Pneumonia”.

Comment 2: The full name of abbreviations should be provided at first mention in all text including abstract.

Response 2: The changes were made, as suggested.

Comment 3: In Table 1, Male and Female should be written instead of Sex M and Sex F, respectively.

Response 3: The changes were made, as suggested.

Comment 4: The genus names of the bacteria should be abbreviated after first use.

Response 4: The changes were made throughout the manuscript, as suggested.

 Comment 5: No need to mention the aim of the study in Discussion.

Response 5: This paragraph was removed from the Discussion section, as suggested.

Comment 6: The text should be read carefully to correct the typos.

Response 6: The text was carefully read and the typos were corrected throughout.
